# CONSERVING FITNESS EVALUATION IN EVOLUTIONARY ALGORITHMS WITH REINFORCEMENT LEARNING

## ABSTRACT

Evolutionary Algorithms (EAs) have been successfully used for many applications, but the randomness in application of mutation and recombination operators implies that a large number of offspring are of relatively low fitness, and those of high fitness substantially resemble already visited candidate solutions; this is manifested in very little improvement in solution quality after the early generations of execution of an EA. We address this issue in EAs by the proposed "Evolutionary Algorithm using Reinforcement Learning (EARL)" that improves efficiency by imposing constraints on the generated offspring. The proposed approach integrates an actor-critic reinforcement learning agent into the EA pipeline. The agent is trained with a rich state representation that captures both local parental information and global population statistics. A carefully designed multi-component reward function balances four objectives: improving fitness over parents, achieving high population rank, maintaining diversity and optimizing fitness. The agent is optimized with "simple policy optimization," a recent RL algorithm, ensuring both learning stability and exploration. Experimental evaluations on four benchmark problems show that EARL achieves faster convergence, superior best fitness values, and greater population diversity compared to a standard EA. We also evaluate EARL on the real-world application of adversarial object generation for robotic grasp training. Our results demonstrate that EARL can transform evolutionary search into a more directed and efficient optimization approach, with practical implications for domains where fitness evaluation is expensive.

## 1 INTRODUCTION

Evolutionary Algorithms (EAs) are population-based metaheuristic optimization techniques inspired by Darwinian natural selection. By iteratively applying selection, crossover, and mutation, they can explore complex search spaces without relying on gradient information. This flexibility has led to successful applications in engineering design (Deb & Gulati, 2001), machine learning (Real et al., 2017), and robotics (Sims, 2023).

A crucial step in EAs is evaluating the fitness of individuals in the population. However, in many application areas, fitness evaluation tends to be a computationally expensive bottleneck. When random EA operators produce ineffective individuals – which happens frequently – the result is not only a poor candidate but also wasted computational resources.

Our goal is therefore to design an EA framework that:

- reduces wasted offspring by biasing variation toward productive candidates,
- maintains diversity and exploration so as not to collapse prematurely, and
- scales to expensive real-world domains, where efficiency is critical.

To this end, we propose an *Evolutionary Algorithm with Reinforcement Learning (EARL)*, which replaces random variation with learned policies for generating children. By integrating an actor-critic reinforcement learning agent into the EA pipeline, EARL transforms offspring generation into an adaptive process that leverages both parent-level and population-level context, making evolutionary search more directed and efficient. More broadly, EARL makes evolutionary algorithms practical in

domains where brute-force randomness is infeasible. Our empirical results demonstrate the generality and benefits of EARL, by evaluation on several common EA benchmarks as well as the specific application of adversarial object generation for robotic grasping algorithms.

## 2 BACKGROUND AND RELATED WORK

Evolutionary Algorithms (EAs) are optimization procedures inspired by Darwinian evolution. They iteratively apply computational analogues of selection, crossover, and mutation to large populations of candidate solutions, which enables exploration of large search spaces without requiring gradient information. One of the most compelling early demonstrations of EA-driven creativity appeared in evolutionary robotics. Sims (1994) evolved both morphology and neural control for virtual creatures capable of walking and jumping, showcasing the ability of EAs to generate lifelike, autonomous systems through simulated evolution. Beyond robotics, EAs have been widely applied across domains. For example, Wang et al. (2023) combined manifold learning with EAs for cancer gene selection, reducing dimensionality while preserving classification accuracy. In reinforcement learning (RL), Brzęk et al. (2025) showed that exploration-driven EA variants can accelerate hyperparameter optimization for Deep Q-Learning.

Despite these successes, EAs often suffer from inefficiency. Their stochastic operators produce many offspring that are either worse than their parents or redundant with existing individuals. This is tolerable in inexpensive benchmark problems but becomes a critical limitation when evaluating each candidate is costly, as in robotics and engineering design.

To address this challenge, researchers have introduced adaptive EAs that integrate reinforcement learning (RL). RL provides a natural framework for adaptation, since it enables decision-making based on performance feedback. Early efforts applied RL to parameter control (Sakurai et al., 2010), and later Xu et al. (2021) dynamically adjusted mutation and crossover rates with methods such as Q-learning . Another line of work by Fialho et al. (2010) focused on adaptive operator selection, where RL framed the choice of genetic operators as a multi-armed bandit problem, improving robustness by selecting the most effective operator at each evolutionary stage.

More recently, RL has been used to guide offspring generation. Instead of relying solely on random crossover or fixed mutation rates, RL agents learn policies conditioned on parent genomes and population statistics. For example, Irmouli et al. (2023) proposed an EA enhanced with deep RL for flow-shop scheduling, where a neural agent adjusted selection probabilities and mutation rates to produce more effective offspring while preserving diversity. Dong et al. (2024) applied RL to wind-farm layout optimization, dynamically tuning EA hyperparameters to accelerate convergence on multimodal landscapes.

Li et al. (2024) surveyed hybrid algorithms that combine EAs and RL, categorizing them into EA-assisted RL, RL-assisted EA, and fully synergistic approaches. Their taxonomy clarifies where our work lies: EARL is an instance of RL-assisted EA, but unlike most surveyed approaches, it uses an RL agent directly in the variation step to generate offspring. In contrast with earlier methods that primarily focused on parameter control or operator selection, our EA with RL-guidance (EARL) observes both parent-level and population-level states and learns a policy for generating children directly. A multi-component reward balances improvement, ranking, diversity, and fitness, transforming the EA from a purely stochastic process into a semi-directed optimization approach. Furthermore, unlike much related work, we evaluate EARL on common EA benchmarks as well as a specific application, viz., adversarial object generation for robotic grasping.

## 3 METHODOLOGY

Traditional EAs work by maintaining a population of candidate solutions and repeatedly applying selection, crossover, and mutation to generate new children. While effective, this process wastes computational effort because many children either regress in quality or contribute little to population diversity. To address this inefficiency, our EARL framework replaces the random child-generation mechanism with a learned policy, as discussed in the rest of this section.

### 3.1 STATE REPRESENTATION

The reinforcement learning agent receives as input a comprehensive state that captures both local and global information:

- Parent genomes (numerical representation of two selected individuals).

- Fitness of both parents as computed by the objective function.

- Population-level statistics: best fitness, worst fitness, mean fitness, and standard deviation.

This design ensures that the agent can make informed decisions by considering not only the selected parents but also the broader evolutionary context.

### 3.2 POLICY AND VALUE NETWORKS

The policy network is modeled as a Gaussian actor with separate heads for the mean and variance, ensuring flexible exploration of the search space. The value network estimates the expected return, guiding the policy updates. Both networks use fully connected layers with nonlinear activations and dropout for robustness. A representation of Policy and Value networks is shown in Figure 1. It is important to note that for the benchmark problems, each offspring evaluation is a single-step episode with deterministic fitness, so the advantage estimate reduces directly to the observed reward (no bootstrapping is needed). We nevertheless retain the actor–critic structure for consistency across domains and for variance reduction in policy updates. In contrast, for adversarial object generation, where object placement and simulation introduce stochasticity, we employ critic-based bootstrapped returns to stabilize learning under noisy rewards. We have also varied the number of hidden layers in each network ranging from 3 hidden layers (including the mean and std dev heads, a single hidden layer in the shared layer) to 7 layers as shown in the image.

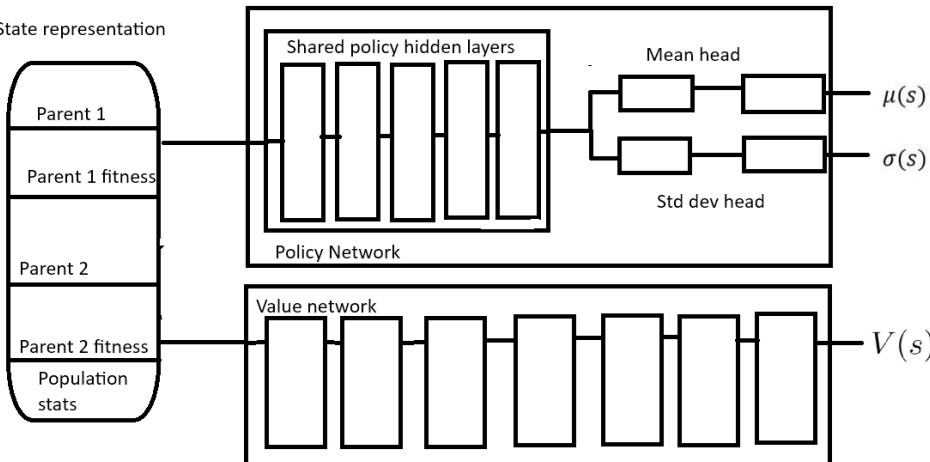

Figure 1: EARL architecture and flow; each box represents a single linear layer followed by non-linear element-wise activation function (ReLU).

### 3.3 REWARD FUNCTION

The reward encourages the agent to propose children that are simultaneously better, competitive, and diverse. It is a sum of the following four components:

- **Improvement Reward:** Measures relative improvement of the child compared to the better parent.

Let $f(\cdot)$ denote fitness, where lower values are better. If $p_1$ and $p_2$ are the parents and $c$ is the child, we define

$$f_{\text{best}} = \min\{f(p_1), f(p_2)\}, \quad R_{\text{improve}} = \begin{cases} \dfrac{f_{\text{best}} - f(c)}{f_{\text{best}} + \epsilon}, & f_{\text{best}} > 0 \\ \max(0, -f(c)), & f_{\text{best}} = 0, \end{cases}$$

where $\epsilon$ is a small constant to avoid division by zero (Ackley's global minimum is at 0). The second branch handles the corner case when the best parent has fitness 0 to avoid division by zero. For functions like Ackley, where fitness is always non-negative, this term is always zero.

- **Ranking Reward:** Reflects how the child ranks compared to the rest of the population. If $P_g$ is the set of all children in generation $g$, then

$$R_{\text{rank}} = \frac{1}{|P_g|} \sum_{x \in P_g} \mathbf{1}[f(c) < f(x)] - 0.5,$$

where the subtraction recenters the reward around zero.

- **Diversity Reward:** Rewards children that are distant from existing individuals, preventing premature convergence; preferring higher distance to the nearest neighbour.

$$d_{\text{min}} = \min_{p \in P_{g-1}} \|c - p\|_2, \quad R_{\text{div}} = \min\left(\frac{d_{\text{min}}}{\alpha}, 1.0\right),$$

where $\alpha$ scales the distance for normalization (problem specific, for example: set to 10.0 in Ackley).

- **Fitness Reward:** A direct copy of the fitness value, to promote optimization of the children.

$$R_{\text{fit}} = -f(c).$$

Each component addresses a different failure mode of evolutionary search: stagnation, loss of competition, lack of diversity, and insufficient fitness pressure. To balance these objectives, we assign equal weights (1.0) to all terms. This simple scheme proved effective across the benchmark problems, where no single objective consistently dominated. While domain-specific fine-tuning of weights could yield further gains in specialized applications (e.g., real-world robotics tasks), we found equal weighting to provide a robust and general-purpose default. We also made two adjustments for the Rosenbrock function only, to account for its large range of fitness values: We replaced $f$ with its logarithm (base 2) in the foregoing formulas, and passed improvement and fitness rewards through tanh to keep them bounded.

## 3.4 LEARNING ALGORITHM

The RL agent is trained using Simple Policy Optimization (SPO) (Xie et al., 2025), a recent first-order reinforcement learning method that addresses limitations of both PPO and TRPO. Traditional PPO uses a clipped surrogate objective that can introduce bias and lead to premature convergence, while TRPO enforces a hard trust region via second-order optimization, which is computationally expensive. SPO introduces a quadratic penalty on policy updates, achieving stable performance with much lower computational overhead.

**SPO Objective.** The policy is updated by maximizing the following surrogate loss:

$$\mathcal{L}_{\text{SPO}}(\theta) = \mathbb{E}_t\left[\rho_t(\theta)\hat{A}_t - \alpha\left(\rho_t(\theta) - 1\right)^2\right] + \beta\,\mathcal{H}[\pi_\theta(\cdot|s_t)], \tag{1}$$

where $\rho_t(\theta) = \frac{\pi_\theta(a_t|s_t)}{\pi_{\theta_{\text{old}}}(a_t|s_t)}$ is the importance sampling ratio, $\hat{A}_t$ is the critic-based advantage estimate, and $\mathcal{H}$ is the policy entropy. The quadratic penalty ($\alpha$) softly constrains the update size, preventing destabilizing jumps, while entropy regularization ($\beta$) preserves exploration.

**Suitability for EARL.** Fitness-based rewards in evolutionary settings are often noisy, as fitness evaluation may include some stochasticity. In preliminary trials, PPO was prone to instability as the reward components increased, and TRPO was impractical due to its per-update cost. SPO provided

more reliable convergence while maintaining computational efficiency, making it particularly well-suited for embedding into EA pipelines.

**Implementation Details.** Following Xie et al. (2025), we adopt first-order gradient updates with Adam (learning rate $3 \times 10^{-4}$). We set $\alpha = 0.5$ for the quadratic penalty and the entropy weight $\beta$ to $0.01$ .

**Exploration–Exploitation Balance.** The entropy term encourages broad exploration in early generations (useful for multimodal functions such as Rastrigin and Ackley). Without entropy regularization, we observed premature policy collapse, with the RL agent generating near-duplicate offspring.

### 3.5 INTEGRATION INTO EA

We embed the RL-guided child generator into the DEAP EA framework (Fortin et al., 2012) to ensure reproducibility and fair comparison with standard EA pipelines. Each generation begins by preserving a fixed number of elite individuals, guaranteeing that the highest-performing candidates are not lost due to stochastic variation. The remaining offspring are generated through a hybrid mechanism where a proportion is generated by the RL mechanism and the rest are generated by the standard crossover and mutation operators. This hybridization provides a balance between exploration and exploitation. The RL agent exploits its learned policy to bias search toward promising regions, while the stochastic EA operators maintain diversity and reduce the risk of premature convergence. In practice, we find that values of proportion between 0.5 and 0.8 provide stable improvements, though this proportion may be tuned per domain. Additionally, we introduce a diversity safeguard: If the RL agent repeatedly generates near-duplicate offspring, the system defaults to random crossover for those cases. Integration requires no modification to the fitness evaluation pipeline, making the approach compatible with any existing EA setup where DEAP or similar frameworks are used. This modular design allows our method to be plugged into both benchmark test functions and real-world domains (e.g., adversarial object generation), with minimal engineering overhead.

## 4 EXPERIMENTS & RESULTS

To evaluate EARL, we conducted experiments on four well-known benchmark functions: Sphere (simple $\ell^2$ norm minimization), Rastrigin (Rastrigin, 1986), Ackley (Ackley, 1987), and Rosenbrock (Rosenbrock, 1960). These benchmarks were selected because they capture a range of optimization challenges, from simple unimodal landscapes to highly multimodal and deceptive search spaces. We also evaluated EARL in a real-world application to adversarial object generation for robotic graspers.

For the baseline EA, we use a standard real-coded genetic algorithm implemented in DEAP (Fortin et al., 2012). Parent selection is tournament-based, crossover is simulated binary crossover (SBX) (Deb et al., 1995) and mutation is Gaussian with adaptive variance. This configuration is widely used in continuous optimization and provides a strong baseline for comparison. EARL replaces a subset of these variation operations with the learned policy, while preserving the rest of the evolutionary loop (elitism, population update, fitness evaluation). Population size is 100 for all experiments.

### 4.1 SPHERE FUNCTION

The Sphere function is a convex, unimodal objective with a single global optimum at the origin. Since this problem is trivial for most evolutionary algorithms, our aim was not to test optimization difficulty, but to evaluate efficiency when offspring generation is constrained.

In this setup, we deliberately limited the number of children produced per generation, reflecting real-world scenarios where fitness evaluations (e.g., physical robot trials) are a major computational bottleneck. A reinforcement learning agent was trained using the Simple Policy Optimization (SPO) update. During evaluation, the agent guided child generation in place of standard crossover/mutation. The RL agent is allowed to add half as many children as the standard EA. Results show that the RL-guided EA achieves the same convergence rate as the baseline EA, but with significantly fewer wasted offspring. Here, "wasted" means any offspring that is created in a generation and does not survive to the next generation. This confirms that at least on a trivial landscape, RL guidance im-

proves efficiency by reducing the number of redundant evaluations, which is crucial for applications where evaluating fitness is expensive.

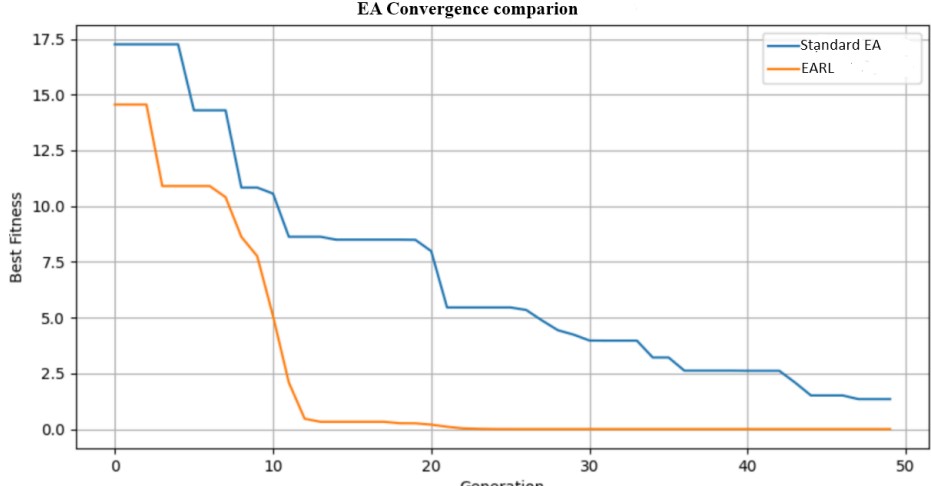

Figure 2: Sphere function: Best fitness over generations for Standard EA vs. EARL.

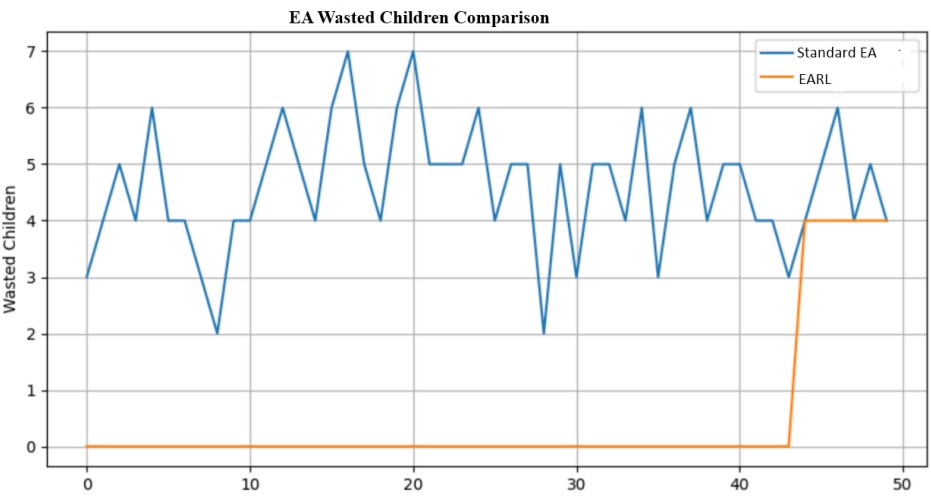

Figure 3: Sphere function: Wasted children over generations for Standard EA vs. EARL.

### 4.2 RASTRIGIN FUNCTION

Rastrigin is multimodal, with many local minima that easily trap evolutionary algorithms. EARL significantly outperforms EA by generating offspring that escape local optima. Thanks to the ranking and diversity terms in the reward, EARL maintains exploration while steadily improving best fitness. No self fitness is included in the rewards since the search space is small (-5,5); the RL agent is able to escape local minima, utilizing the ranking, diversity, and improvement components of the reward.

To further explore the stability of SPO in our setting, we also tested actor–critic architectures of different depths, ranging from a lightweight 3-layer configuration to a deeper 7-layer network. Prior work like Moalla et al. (2024) and Dohare et al. (2024) has noted that PPO-based updates can become unstable for deep policies, often leading to premature collapse or poor convergence. In contrast, the recent SPO algorithm has been shown to support stable optimization with deeper ar-

chitectures. Our results on the Rastrigin benchmark (Figures 4 and 5 ) confirm this: SPO enables EARL to maintain robust learning even with a 7-layer network.

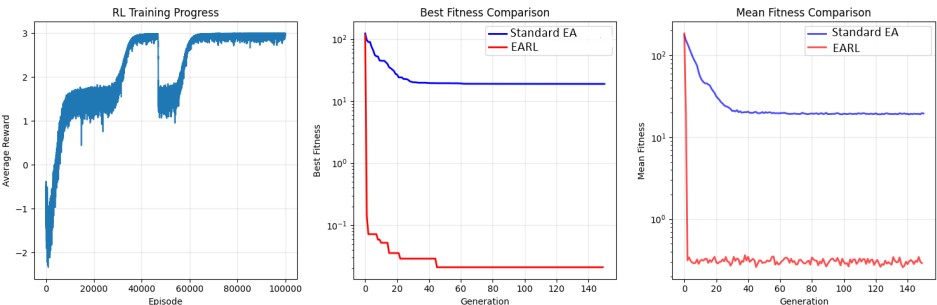

Figure 4: Rastrigin function: 7-layer Network Convergence comparison of Standard EA vs. EARL.

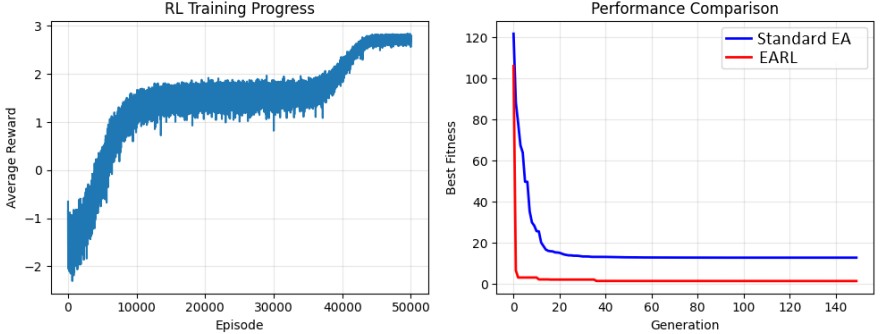

Figure 5: Rastrigin function: 3-layer Network Convergence comparison of Standard EA vs. EARL.

### 4.3 ACKLEY FUNCTION

Ackley features wide flat plateaus with numerous local minima, making it difficult for random operators to progress. Standard EA often stagnates. In contrast, EARL learns strategies to generate children that cross flat regions and continue toward better solutions (Figure 6). This is achieved by balancing diversity with improvement incentives. We include the self fitness reward component here since the search space is large with wide plateaus, and notice that this helps the agent escape the wide flat regions.

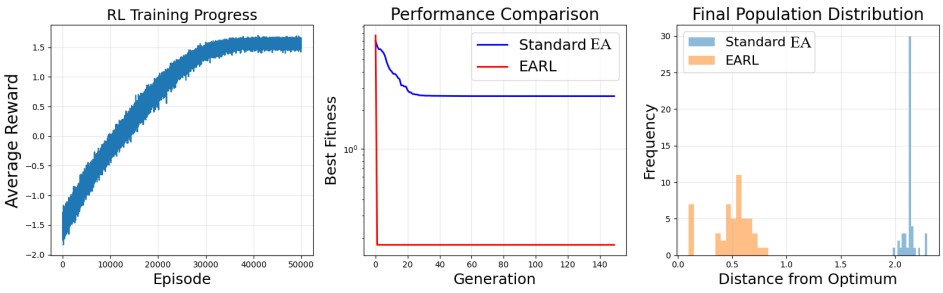

Figure 6: EARL and Standard EA performance on the Ackley function.

## 4.4 ROSENBROCK FUNCTION

Rosenbrock presents a curved valley where naive crossover leads to oscillations. Standard EA struggles to follow the valley's narrow path. EARL produces smoother child trajectories, guiding the search along the valley floor and achieving faster convergence (Figure 7).

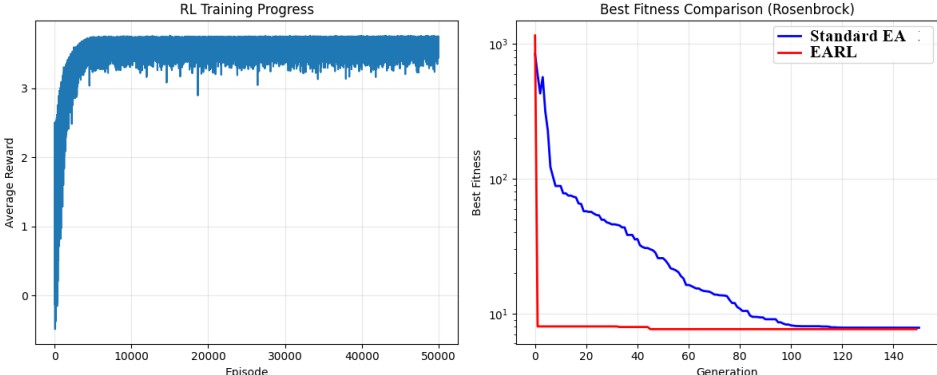

Figure 7: Rosenbrock function: Best fitness progression showing EARL vs Standard EA

## 4.5 DISCUSSION OF TRAINING COSTS

One might observe that the reported training curves for EARL show up to 50k agent episodes, which appears larger than the 14k evaluations used by a standard GA (population size 100, 140 generations). However, this comparison is misleading for three reasons. First, the RL agent typically converges far earlier—within $\sim$ 10k episodes in most of our benchmark experiments—and the remaining episodes simply illustrate training stability. Second, on multimodal functions such as Rastrigin and Ackley, standard GA never achieves the same best fitness as EARL, even after its evaluations are exhausted, whereas EARL continues to improve. Finally, the policy learned by EARL can be reused across multiple independent EA runs or transferred to related problems, so the training cost is amortized rather than repeated each time. Thus, although the raw episode count is higher, EARL offers more reliable convergence to superior solutions under equivalent or smaller effective budgets.

## 4.6 ADVERSARIAL OBJECT GENERATION

We also evaluate EARL on adversarial object generation for robotic grasp testing. Objects are represented in a voxelized form, where each voxel encodes occupancy in a 3D grid. Candidate objects are generated and evolved by modifying voxel configurations through crossover and mutation. Fitness evaluation is performed in a physics simulation, where a simple robotic grasper attempts to grasp at every possible voxel location. The grasper itself is not intelligent and does not plan grasps; instead, it exhaustively tests voxel positions. An object is considered difficult if no stable grasp is found across these attempts. This makes fitness evaluation computationally expensive, since each candidate must be voxelized, simulated, and tested under multiple trials. Each candidate object is voxelized, simulated in a physics engine, and evaluated by a graspability test; consequently, fitness evaluations are expensive and we operate on a population (a list of objects) per generation. Figure 8 plots both the *mean* and *best* fitness across generations for the standard EA and EARL.

It is important to emphasise the evaluation choice: we report and analyze *mean* population fitness because our experimental pipeline is population-driven — the next generation must consist of a more challenging set of objects in aggregate for adversarial training to be effective. In this setting, a single, early low-fitness individual produced by randomness in the baseline EA can be misleading: standard EA occasionally finds a very strong adversarial object quickly due to stochastic variation, but such solutions are outliers and do not imply that the population as a whole has become more adversarial.

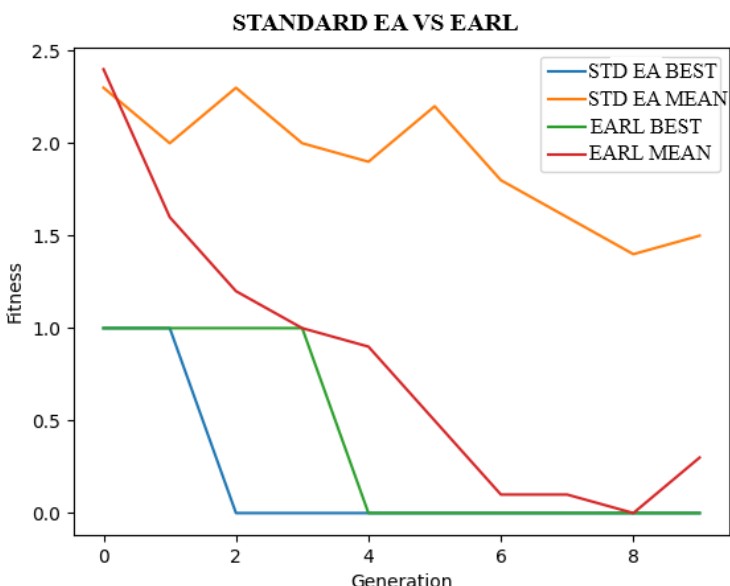

Figure 8: Mean and Best fitness for Adversarial object generation

Our observations from Figure 8 are therefore twofold: (i) The standard EA sometimes attains the best individual earlier than EARL, reflecting lucky, high-variance draws from stochastic operators; (ii) EARL produces a steadier and sustained reduction in *mean* fitness across the population, indicating consistent population-level hardening. This steady mean fitness improvement is attributable to the RL agent biasing offspring generation towards diverse, difficult configurations rather than relying on chance. Thus, while the baseline can occasionally produce a single strong adversarial example early, EARL more reliably elevates the difficulty of the entire population — which is the desirable outcome when generating adversarial datasets for robotic grasp training.

## 5 CONCLUSION

We introduced EARL, an evolutionary algorithm guided by reinforcement learning that integrates an actor-critic agent into the offspring generation process. By leveraging a multi-objective reward and training with Simple Policy Optimization, EARL biases search toward productive and diverse solutions while reducing wasted evaluations.

Experiments on benchmark functions demonstrate that EARL improves efficiency on trivial landscapes (Sphere), avoids local minima (Rastrigin), escapes flat plateaus (Ackley), and smoothly navigates curved valleys (Rosenbrock). In the real-world task of adversarial object generation, EARL steadily decreases mean population fitness, producing consistently harder object sets for robotic grasp training. While standard EA can occasionally find a strong adversarial object by chance, EARL provides more reliable population-level improvement — a property critical in expensive,population-driven domains.

There are several promising directions for extending EARL. On the application side, EARL could be embedded in co-evolutionary setups where both the optimization environment and the agent adapt simultaneously, such as adversarial training with robotic graspers or other manipulation tasks. More generally, EARL could benefit from advances in adaptive reward weighting, richer state encodings (e.g., graph-based or learned embeddings of individuals), and transfer learning across problem domains. Exploring these enhancements may yield even greater efficiency and robustness, further establishing EARL as a practical tool for optimization under expensive fitness evaluation.

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
