# OpenReview forum: "Conserving Fitness Evaluation in Evolutionary Algorithms with Reinforcement Learning"
_ICLR.cc/2026/Conference — Submitted to ICLR 2026_

### Official Review · Reviewer_aDwx · 2025-10-23

**Soundness:** 2
**Presentation:** 3
**Contribution:** 2
**Rating:** 2
**Confidence:** 3

**Summary:**

The manuscript introduces EARL, an evolutionary algorithm in which an actor–critic RL agent replaces part of the stochastic variation step, crossover and mutation to generate offspring more “productively.”

The agent observes parent genomes and fitness, plus population-level statistics (best/worst/mean/std), and outputs parameters for generating children; rewards combine relative improvement over parents, population rank, diversity in the form of nearest-neighbour distance, and raw fitness, typically with equal weights. Training uses Simple Policy Optimization (SPO) with entropy regularization.

The method is integrated into DEAP, with elitism and a hybrid fraction of RL and EA-generated offspring. Experiments cover Sphere, Rastrigin, Ackley, Rosenbrock and a robotic grasping “adversarial object generation” task.

**Strengths:**

1. Originality. The authors' offspring generation as a contextual policy that directly outputs children is a clear, focused instantiation of “RL-assisted EA". This is an approach distinct from prior parameter control or operator selection, e.g., bandit-style adaptive operator selection.

2. Following up on (1) above. The reward decomposition is sensible and maps onto known EA failure modes, namely stagnation, mode collapse, and weak fitness pressure.

3. The manuscript clearly enumerates state features, the four reward terms stating their equations, and the high-level training loop. Figures in the paper show fitness vs. generations for each benchmark and the application task.

**Weaknesses:**

1. Figures appear single-seed or under-reported; there are no confidence intervals, number of seeds, or variance beyond narrative statements. The adversarial-objects experiment emphasizes that mean fitness is acceptable (good) but still requires multiple seeds, effect sizes, and robustness analyses with respect to noise sensitivity and simulation randomization.

2.  There are reproducibility gaps. In my view, essential details are missing. For example, exact dimensionality, search ranges per function, SBX parameters, mutation variance schedule, elitism count, selection pressure, episode batching, learning-rate/entropy schedules, and stopping criteria.

3. In my view, comparisons to canonical “directed variation” ideas are missing. The authors claim that their EARL is moving from “purely stochastic” to “semi-directed” search, but established ES variants already implement directed search via gradient-like updates of a search distribution. Examples include, but are not limited to,  NES, CMA-ES or QD algorithms, which explicitly trade off quality and diversity. A head-to-head comparison and inclusion of the results would greatly improve the utility of this manuscript.

4. I do not wish to be overly negative, but one may claim that the benchmarking protocol is below expected standards. The Sphere/Rastrigin/Ackley/Rosenbrock set is too thin. Moreover, it is under-specified, for example, with no explicit dimensionalities, instance variations, termination criteria, or evaluation budgets beyond population size. It can be claimed that modern black-box optimization studies rely on COCO/BBOB with prescribed dimensions and multiple instances to obtain fair, reproducible comparisons and (meaningful) statistical tests.

5. It can be claimed that the reward design and credit assignment issues reduce interpretability. Some statements (and possible questions for the authors) with respect to this claim follow below:

• The ranking reward depends on the full set of children in the same generation; this non-stationary target can destabilize learning and couple an individual’s reward to unrelated samples.
• The diversity reward uses raw Euclidean distance with a problem-dependent scaling constant (α); this is not scale-invariant across domains and dimensions and creates a tuning burden.
• Equal weighting of the four terms is asserted to be “effective,” but no ablations are shown to justify this choice or to quantify sensitivity.

A reasonable fix is to provide the missing ablation studies.

**Questions:**

See commentary above (especially weaknesses identified in (3)-(5)).

In its current form, the manuscript presents a promising engineering idea but lacks comparisons with state-of-the-art optimizers and COCO-style benchmarking. I also recommend that a careful ablation and statistical analysis be introduced.

---

### Official Review · Reviewer_36k3 · 2025-10-29

**Soundness:** 2
**Presentation:** 1
**Contribution:** 2
**Rating:** 4
**Confidence:** 4

**Summary:**

This paper embeds an actor-critic reinforcement learning agent into the offspring generation step of an evolutionary algorithm. The learned policy replaces or constrains standard crossover and mutation to reduce low-quality or redundant offspring. The state input includes parent genotypes and basic population statistics. The reward has four parts: relative improvement, population ranking, diversity, and fitness. The policy is trained with Simple Policy Optimization. Experiments on four continuous benchmarks and a robotic object-generation task report faster convergence, better best and mean fitness, and fewer wasted offspring.

**Strengths:**

- The paper focuses on a meaningful problem: expensive fitness evaluations and wasted offspring from blind variation.
- Using RL to assist EA is timely and interesting.

**Weaknesses:**

- The paper’s structure has some issues. In the Methods section, the overall workflow is not clearly explained, and it remains unclear how RL and EA actually interact. I suggest that the authors first present this key information (a clear overall flow), and then describe RL and EA separately. In addition, the RL state, action, reward, and termination conditions should be explicitly defined.

- The experimental results seem to lack statistical analysis. The figures only show single curves without shaded error bands or variance information.

- As I understand it, RL directly generates solutions while EA searches in the solution space, with each responsible for about half of the offspring. Since RL and EA interact in this hybrid design, how does it compare with maintaining an independent RL process? Recent works such as ERL-Re2 and EvoRainbow have shown that collaborative optimization often performs better than EA-dominated schemes, and they provide supporting experiments. Including a deeper analysis and comparison in this aspect would significantly enhance the overall contribution and clarity of the paper.

- The experimental environments are overly simple, and the chosen baselines are relatively weak.

**Questions:**

- How does the proposed method compare with pure RL method (single agent) applied to the same tasks?

- How are the relative weights among the four reward components chosen and balanced in practice?

- Given the inclusion of an explicit diversity term, how does the method relate to Quality-Diversity approaches, and can the authors comment on similarities or differences?

---

### Official Review · Reviewer_zrMp · 2025-10-31

**Soundness:** 2
**Presentation:** 2
**Contribution:** 3
**Rating:** 2
**Confidence:** 4

**Summary:**

The paper proposes a new algorithm, Evolutionary Algorithm using Reinforcement Learning (EARL), which uses RL to learn a conditional distribution from which offspring are sampled during evolutionary search, rather than by using the standard operators typically used in evolutionary algorithms.
EARL is evaluated on four standard optimisation benchmarks and one real-world application domain - adversarial object generation.
The results show that on the optimisation benchmarks EARL significantly outperforms a baseline EA in terms of best fitness and wasted children.
On the adversarial object generation task EARL is shown to outperform the baseline with respect to the mean population fitness through time.

**Strengths:**

1. The main argument on which the paper is built, that naive mutations in EAs can produce many low fitness and uninformative offspring is certainly true, and is currently an important problem to tackle for the community.
2. Considerable performance improvements over the baseline are demonstrated on multiple experiments.
3. A wide range of domains are considered, consisting of both standard optimisation benchmarks and a real-world application.

**Weaknesses:**

1. It is unclear whether the RL agent is trained prior to evolutionary optimisation and kept static, or whether it is trained in conjunction with the evolutionary optimisation process. If the former, then the fitness space must be first traversed in order to generate the training data to train the agent, and each of those training points must be evaluated for their fitness greatly increasing the number of fitness evaluations, which could be costly. Therefore, the solution essentially has to be located prior to the evaluation, making the subsequent evaluation essentially redundant. I am also concerned that if the training of the agent is done beforehand then the agent essentially memorises the fitness space and that this is the main reason for the impressive performance. If the agent is trained in conjunction with the evolutionary optimisation procedure, then its impressive performance in the early generations is especially surprising to me but if valid are very impressive results.
2. The paper claims that the performance of EARL is down to the diversity aspect of the reward function, but given that the approach is hybrid it could be in part, or fully, due to the standard operators instead. I am confused as to why there is no comparison to a non-hybrid approach that solely uses an RL agent for offspring production because this would have helped to discern where the improvements could be attributed.
3. Some of the important details of the experiments are missing, such as how the RL agent is trained, in what way the hidden layers in the policy network are shared, whether the offspring are in fact sampled from the Gaussian distribution output by the policy network, whether the proportion of RL to EA is calculated as EA/(EA+RL) or RL/(EA+RL), how wasted children is calculated.
4. There is no information on whether the hyperparameters of the baseline EA were optimised, if not the baseline may have been unfairly evaluated.
5. It is unclear how this algorithm differs compared to other algorithms already proposed in the literature.
6. Experiments are not repeated.
7. Images are not vector images and as a result are pixelated. Some of the images are rendered with mismatching fonts and with poor quality. It would have been more clear if Figures 4 and 5 were combined.
8. Safe Mutations for Deep and Recurrent Neural Networks through Output Gradients (Lehman et al. 2018) was not cited, but I think it should have been included in related work due to its relevance.

**Questions:**

1. How is the training data generated to train the RL agent? Is the RL agent trained prior to the evaluation and if so is it kept static during the evaluation? If it is trained in conjunction with the evaluation this would result in me increasing my overall rating.
2. Why are the results of a non-hybrid approach not reported?
3. Which specific previous work is most like this work and how does this work differ? Is this the first time that parent-level _and_ population-level states are combined in the full state and is the proposed reward function novel? The last paragraph in section 2 refers to this but I would like the novelty to be highlighted and more specific, currently it feels a bit hand wavy. What specifically is different compared to what has been done before?
4. Were experiments with a richer state representation ran? It is surprising to me that the state does not need to be richer than it is in order to generate such impressive results.

---

### Official Review · Reviewer_4BPj · 2025-11-02

**Soundness:** 1
**Presentation:** 1
**Contribution:** 1
**Rating:** 0
**Confidence:** 5

**Summary:**

The motivation is to resolve this inefficiency by reducing wasted computational resources from ineffective offspring, maintaining population diversity to avoid premature convergence, and enabling EAs to scale to real-world domains where fitness evaluation is costly.
The core method proposed is called EARL (Evolutionary Algorithm using Reinforcement Learning), which integrates an actor-critic RL agent into the EA pipeline, a multi-component reward function (balancing fitness improvement, population ranking, diversity, and direct fitness), and optimizing the agent via the Simple Policy Optimization (SPO) algorithm.
Experimental results demonstrate that EARL outperforms standard EAs on four benchmark functions (Sphere, Rastrigin, Ackley, Rosenbrock) in terms of faster convergence, superior best fitness, and greater population diversity.

**Strengths:**

The four-component reward function (Section 3.3) explicitly targets key EA failure modes (stagnation, lost competition, insufficient diversity, weak fitness pressure), which make sense to me.

**Weaknesses:**

0. EARL is not a proper name for the proposed method.

1. The introduction part is overly simplistic.

2. The motivation is not verified.

3. The quality of all the figures should be greatly improved. Please refer to the figures from the accepted papers from top-tier conferences.

4. Regarding the equal reward weights (Section 3.3, line 136: “To balance these objectives, we assign equal weights (1.0) to all terms”):

5. Comparison of SPO vs. other RL methods is lacking.

6. Experiments is not comprehensive: lacking important benchmarks (e.g., more real-world black-box optimization tasks) and state-of-the-art learning-based black-box optimization algorithms.

**Questions:**

See weaknesses

---

### Meta-Review · Area_Chair_rJcZ · 2025-12-29

**Summary:**

The paper proposed a new method leveraging evolutionary learning and reinforcement learning. The reviewers noted significant issues with the presentation of the paper and evaluation of the proposed method with multiple reviewers noting subpar practices. The authors appear to not have participated in the discussion period to address any of the raised concerns and in the current state none of the reviewers support the paper.

**Reviewer Concerns:**

The authors did not address any reviewer concerns.

**Reviewer Scores:**

The authors did not address any reviewer concerns and the scores would remain the same.

---

### Decision · Program_Chairs · 2026-01-26

Reject